# Transfers of Care between Healthcare Professionals in Obstetric Units of Different Sizes across Spain and in a Hospital in Ireland: The MidconBirth Study

**DOI:** 10.3390/ijerph17228394

**Published:** 2020-11-13

**Authors:** Anna Martín-Arribas, Rafael Vila-Candel, Rhona O’Connell, Martina Dillon, Inmaculada Vila-Bellido, M. Ángeles Beneyto, Inmaculada De Molina-Fernández, Nerea Rodríguez-Conesa, Cristina González-Blázquez, Ramón Escuriet

**Affiliations:** 1GHenderS Research Group, School of Health Sciences Blanquerna, Universitat Ramon Llull, Carrer Padilla 326, 08025 Barcelona, Spain; annama7@blanquerna.url.edu (A.M.-A.); rescuriet@gencat.cat (R.E.); 2Faculty of Medicine, Universidad Autónoma de Madrid, Calle Arzobispo Morcillo 4, 28029 Madrid, Spain; cristina.gonzalez01@uam.es; 3La Ribera Hospital Health Department, Carretera Corbera km 1, 46600 Valencia, Spain; 4Foundation for the Promotion of Health and Biomedical Research in the Valencian Region (FISABIO), 46020 Valencia, Spain; 5Faculty of Nursing and Podiatry, Universitat de València, Jaume Roig, s/n, 46010 Valencia, Spain; 6School of Nursing and Midwifery, University College Cork, T12 YN60 Cork, Ireland; r.oconnell@ucc.ie; 7Cork University Maternity Hospital, Wilton, T12 YE02 Cork, Ireland; martina.dillon@hse.ie; 8Verge dels Lliris Hospital, Poligon de Caramanxel s/n, 03804 Alcoi, Spain; vila_inmbel@gva.es (I.V.-B.); angelesbeneytofrances@gmail.com (M.Á.B.); 9Nursing Department, Universitat Rovira i Virgili, 43003 Tarragona, Spain; inmaculada.demolina@urv.cat; 10Rio Hortega Hospital, Calle Dulzaina, 2, 47012 Valladolid, Spain; nrodriguezc@saludcastillayleon.es; 11Catalan Health Service, Government of Barcelona, 08028 Catalonia, Spain

**Keywords:** midwife-led care, obstetrician-led care, continuity of care, transfer of care, maternal outcomes, neonatal outcomes

## Abstract

Background: In Europe, the majority of healthy women give birth at conventional obstetric units with the assistance of registered midwives. This study examines the relationships between the intrapartum transfer of care (TOC) from midwife to obstetrician-led maternity care, obstetric unit size (OUS) with different degrees of midwifery autonomy, intrapartum interventions and birth outcomes. Methods: A prospective, multicentre, cross-sectional study promoted by the COST Action IS1405 was carried out at eight public hospitals in Spain and Ireland between 2016–2019. The primary outcome was TOC. The secondary outcomes included type of onset of labour, oxytocin stimulation, epidural analgesia, type of birth, episiotomy/perineal injury, postpartum haemorrhage, early initiation of breastfeeding and early skin-to-skin contact. A logistic regression was performed to ascertain the effects of studied co-variables on the likelihood that participants had a TOC; Results: Out of a total of 2,126 low-risk women, those whose intrapartum care was initiated by a midwife (1772) were selected. There were statistically significant differences between TOC and OUS (S1 = 29.0%, S2 = 44.0%, S3 = 52.9%, S4 = 30.2%, *p* < 0.001). Statistically differences between OUS and onset of labour, oxytocin stimulation, type of birth and episiotomy or perineal injury were observed (*p* = 0.009, *p* < 0.001, *p* < 0.001, *p* < 0.001 respectively); Conclusions: Findings suggest that the model of care and OUS have a significant effect on the prevalence of intrapartum TOC and the birth outcomes. Future research should examine how models of care differ as a function of the OUS in a hospital, as well as the cost-effectiveness for the health care system.

## 1. Background

The past few years have witnessed the emergence of a worldwide debate about the care that healthy women receive when giving birth. This debate includes differing opinions as to the use of various technologies for normal or low-risk procedures [1], the concept of labour as a physiological process that is not to be understood in solely medical terms, and the growing role women themselves are taking in the decision-making processes affecting their care [2]. Additionally, the increase in the frequency of interventions carried out during labour, especially of caesarean sections, is a source of concern for a number of organization and for health care officials. In different European countries, the rates are about 25% to 35% [3,4].

The organizational structure of maternity services might have an influence on the health outcomes of women and new-borns [5]. The way these medical services are organized determines the level of continuity of care offered and the choice of which professionals provide what kinds of care throughout the process of pregnancy, labour and postpartum care. A number of studies have shown positive results when midwives are the primary providers of care throughout the process [6].

Maternity care in Europe is offered by organizations with a number of different kinds of structures, settings and locations [7]. The degree to which European health care systems offer coverage varies from country to country [8]. In some places, women are guaranteed coverage for the whole range options, whether women choose home births, birth centres, maternity units attended by midwives or conventional obstetric units. The size of maternity units also vary as does the degree of midwifery autonomy [6]. The most common model in European countries is that of providing care in conventional obstetrics units [9]. This translates into a situation in which most healthy women receive care in highly technological settings. Additionally, the frequency of intrapartum interventions varies greatly from place to place [10].

The quality of care provided to women during labour has been extensively studied by a number of researchers [11]. Nonetheless, most of the indicators used in these studies have been aimed at assessing the use of interventions and at seeking out results based on pathology (i.e., postpartum haemorrhages, perineal tears, etc.). The aim of the MidconBirth study is to offer a new perspective on this issue and to contribute to research on the assessment of the quality of care that women receive when in labour [9]. In most health care services, midwives are the primary providers of care throughout the process, but this is not always the case [6]. The roles of different care providers vary depending on how health services and medical teams are organized [12]. Most women who give birth are healthy or at least at low-risk during their pregnancies, and they tend to receive care from midwives who exercise complete autonomy from the onset of labour [13]. The transfer of care (TOC) during labour means that midwives are responsible for detecting any risk, problem or pathology requiring the intervention of another professional [11]. In some cases, however, midwives’ degree of autonomy is affected by the organizational structure of professional teams. In Spain, most women in labour receive care in obstetric units staffed by both obstetricians and midwives [14]. These units have all the necessary technology to provide care for women regardless of the level of risk present in the pregnancy. Care is provided by professional teams organized into hierarchical structures [7]. As a result, procedures are often determined by protocols, and professionals tend to work in accordance with a given centre’s organizational culture. In general, midwives tend to be responsible for women with low-risk pregnancies during labour [6]. The degree of autonomy exercised by these midwives varies from centre to centre [6]. Meanwhile, obstetricians act as consultants in these cases. In Ireland, intrapartum care for women with low-risk pregnancies is provided under the Mother and Infant Care Scheme [15], a program that promotes individualized care for low risk women in labour by midwives and under which obstetricians act only upon the request of these midwives.

Studies have shown that midwife-led models are associated with both fewer medical interventions and increased satisfaction with the birthing experience [6]. However, a recent study on obstetric interventions in Spain suggests the need for further examination of factors associated with the organisation of childbirth services which are influencing these interventions [13].

This study examines the relationships between the intrapartum transfer of care from midwife to obstetrician-led maternity care and the obstetric unit size (OUS), intrapartum interventions and birth outcomes within two different countries with a different midwifery organisation of care.

## 2. Methods

This article presents part of the results obtained within the MidconBirth study. This is a prospective, multicentre, cross-sectional study promoted by the COST Action IS1405 carried out in different hospitals in Spain and Ireland. The protocol can be accessed through the registry ISRCTN14062994 [9]. For the purposes of this study, we selected births attended to in obstetric units in three different regions that are representative for Spain in terms of sociodemographic and economical characteristics (Catalonia, C. Valenciana and Castilla y León), and another in Ireland. These include hospitals with low volumes of births (<600 births per year) or Unit Size 1 (S1), medium (from 601 to 1200 births) and high annual volumes of births (1201 to 2400 births) or Unit Size (S2) and Unit Size (S3) respectively, as well as births attended to by a continuity of care team in Ireland (Cork), a hospital with high annual volume of births (>2400 births) or Unit Size 4 (S4). Data were collected through an online platform in 2016–2019. The sample was limited to primiparous and multiparous women between 18 and 40 years of age with a singleton, cephalic presentation and uncomplicated pregnancy between 37 and 42 weeks of gestation. For this study, women with pregnancies classified as high or very high risk were excluded.

The reference population was 5708 women. The sample size is calculated on the annual number of births of each participating centre or midwife. To calculate the sample size (95% level of confidence) it is assumed an unknown proportion of births attended by midwives for each estimated population (50%) in each setting, with a (+/−) 5% precision and a reposition proportion of 10%. A minimum estimated sample size was 365 women to achieve a representative sample for each hospital in Spain and for the caseload midwifery team in Ireland. Data collection was conducted consecutively during the specified period until the minimum number of cases needed was reached. The primary outcome was transfer of care (TOC). This happens when the professional who is looking after the woman at the start of her labour transfers the responsibility of care to another professional. The secondary outcomes included type of onset of labour (spontaneous or induced labour), oxytocin stimulation (use of oxytocin during the first or second stage of labour), epidural analgesia (use of epidural analgesia during the second or third stage of labour), type of birth (normal or dystocic), episiotomy/perineal injury (the use of episiotomy and/or presence or perineal damage), postpartum haemorrhage (more than 1000 mL of blood loss), early initiation of breastfeeding (within one hour from birth) and early skin-to-skin contact (contact between mother and new-born is started immediately after birth and/or uninterrupted during the first 30 min).

Descriptive statistics were used to summarize the women’s characteristics. The statistical analysis was carried out using the SPSS program version 23.0 (IBM SPSS Statistics for Windows, Version 25.0, released 2018, IBM Corp., Armonk, NY, USA). Frequencies and percentages of the categories were calculated for all the variables. The standard deviation (*SD*) of the quantitative variables mean was calculated. The Chi-square test was used to analyse the statistical significance of the differences in the percentages of hospital groups between the variable categories; for risk factors for transfers of care, an odds ratio (OR) with a 95% CI, was calculated. A multivariate logistic regression models were performed to ascertain the effects of studied co-variables on the likelihood that participants had a TOC. These models were adjusted using a stepwise variable selection process based on a likelihood ratio (LR). Nagelkerke’s R^2^ was used to estimate the coefficient of determination from 0 to 1. The significance level was set at *p* < 0.05.

### Ethics Approval and Consent to Participate

The MidconBirth study was approved by the ethics committee of the coordinating centre (Clinical Research Ethics Committee of Parc Salut Mar 2016/6785/I) ISRCTN registry 17,833,269 and later by the ethics committee of each participating centre (Clinical Research Ethics Committee of the Catalan Hospitals Union Foundation (CPMP/ICH/135/95), Clinical Research Ethics Committee of Rio Ortega Hospital (117/16), Clinical Research Ethics Committee (CREC) Cork Ref ECM4 (09/05/17), Human Ethics Committee at Hospital Universitario de La Ribera Research Ethics Committee and Research Commission and the Spanish Medicines and Medical Devices Agency approved the study (HULR15/12-01), Research Ethics Committee of Complejo Asistencial Universitario de Palencia (CIB-2017005) and Research Ethics Committee of Hospital Verge dels Lliris. Ethics committee approval was required for each participating hospital. Since this is an observational study in which data was anonymized, no consent was required from the women cared for in the participating centres. If a hospital required consent from the women under their care, written consent was obtained. Further information and documentation are available on request.

## 3. Results

### 3.1. Characteristics of the Sample

The total sample analysed was made up of 2126 cases. In terms of the regional distribution, Cork (Ireland) collected data on 7.1% (150) of the cases, 44.7% (951) were in Catalonia, 48.2% (1025) in the regions of Valencia and Castilla y León (Spain). Regarding the obstetric unit size distribution, S1 represented 8.8% (187) of the cases, 51% (1086) were in S2, 33% (703) were in S3 and 7.1% (150) were in S4 (Figure 1).

The average age of the women in the study was 31.7 ± 4.9 years. Broken down by country of origin, 70.2% (1490) of the women were from Spain, 11.8% (250) were from elsewhere in Europe, 7.3% (155) were from South or America, 8.1% (172) were from Africa, 2.0% (42) were from Asia, 0.5% (10) were from the Middle East, 0.2% (four) were from North America and for 0.1% (three) of the women this data was missing. 36.3% (772) of the women had attended university, 34.3% (729) had high school degrees, and 25.2% (535) had only completed primary school, while for 4.2% (90) of the women the level of education was unknown or could not be classified. 52.3% (1111/2126) were primiparous, while the average gestation period before labour was 39.0 ± 3.0 weeks (range of 37–41).

The clinical characteristics of women in every OUS group are shown in Table 1, which also displays the statistically significant differences between OUS, with the exception of the use of epidural anaesthesia (*p* = 0.632).

Of women who had dystocic births, in S1 we observed that 12.3% (23) of women required emergency caesarean sections; in S2 the percentage was 12.6% (137); in S3 16.5% (116) and in S4 11.3% (17). The differences here were statistically significant (*p* < 0.001). S1 displayed a lower percentage of transfer (29.4%), fewer cases of labour stimulated with oxytocin (31.0%) and fewer cases of induced labour (12.4%). Meanwhile, S1 displayed the highest percentages of normal births (77.5%), skin-to-skin contact between the mother and the new-born (96.3%), and early initiation of breastfeeding (88.2%).

S2 showed the highest percentage of induced labour (30.8%), serious perineal injuries including episiotomies and third- and fourth-degree perineal tears (48.9%), but this OUS showed the lowest percentage of postpartum haemorrhages (2.0%).

S3 displayed the greatest percentage of transfer (54.9%), of labour stimulation with oxytocin (70.4%), of the use of epidural analgesics (85.5%) and of dystocic births (37.1%).

S4 showed the lowest proportion of births with epidural analgesia (18.7%), was most likely to lack serious perineal injuries, characterized as cases where the perineum was intact or cases with second and third degree perineal tears (66.7%), and had the highest percentage of the start of labour attended by midwives (99.3%). However, this OUS also displayed the highest percentage of postpartum haemorrhages (7.3%).

### 3.2. Transfer Analysis

We were interested in analysing the relationship between the TOC between the midwife and the obstetrician and the rest of the factors that influence a birth. For the purposes of this analysis, cases of elective caesarean sections (42) were excluded. Thus, the total number of cases analysed was 2084.

Midwives attended the start of the deliveries in 85.1% (1773/2084), and they attended during the expulsive phases of the deliveries in 59.4% (1237/2084) of cases. Meanwhile, obstetricians attended the start of the deliveries in 14.9% (311/2084) of cases, and they attended the end of deliveries in 40.6% (847/2084) of cases.

In 55.5% (1156/2084) of the deliveries, there was no TOC from the midwife to the obstetrician. In other words, in these cases midwives attended the whole labour and birth process. In terms of the distribution by OUS, the midwives in S1 were the least likely to transfer care (with 71.0% [132/186] attending to the labour and birth in its entirety), followed by those in S4 (69.8% [104/149]), S2 (56.0% [1076/673]) and, finally, S3 (47.1% [317/773]).

We conducted an analysis of the differences in the labour and birth processes and the associated perinatal results in each OUS, examining them in terms of whether or not there was a TOC during the process. For the variables analysed, (type of start of labour, pharmacological stimulation of labour, use of epidural analgesics, type of birth and status of the perineum), statistically significant differences were found, both within each obstetric unit size and for the sample as a whole.

It is true that the midwife might not be directly responsible for the decision to induce labour, as this represents a departure from a normal birth because the onset is not spontaneous. However, this process is often determined by protocol and characterized by a shared responsibility of the midwife and the team of obstetricians [7].

Our analysis of the factors associated with a greater likelihood of TOC and the risks associated with this practice is displayed in the 2 × 2 tables and the odds ratio calculations (Table 2).

When there was no TOC, S4 recorded the highest proportion of spontaneous onset of labour [S4 93.3% (97/104), compared with S1 at 90.9% (120/132), S2 at 83.3% (502/603), and S3 at 83.0% (263/317)], and the differences found here were statistically significant (*p* < 0.001). S3 showed the highest rate of induced births attended by midwives in which no TOC occurred [17.0% (54/317), while for S2 the figure was 16.7% (101/603), for S1 it was 9.1% (12/132), and for S4 it was 6.7% (7/104)]. The differences found were statistically significant (*p* < 0.001). S4 displayed the lowest frequency of oxytocin use in deliveries when no TOC occurred [13.5% (14/104)]. Meanwhile, S3 registered the highest rate of pharmacological stimulation (55.5% [176/317]). Additionally, in S4 TOC was more likely when oxytocin was used, or labour was stimulated, increasing with respect to when labour was not stimulated (*p* < 0.001).

In cases where there was no TOC, S3 registered the lowest rate of use of epidural analgesics [75.1% (238/317)], while S1 and S2 displayed the greatest tendency to administer them (78.8% ([104/132] and 78.8% [475/603], respectively). In S1, the risk of TOC was seven times higher when epidural analgesics were administered than when they were not (*p* = 0.003), with the rate reaching 96.3% (52/54) in these cases. The lowest prevalence of TOC associated with the use of epidural analgesics was found in S2 (*p* < 0.001), where the figure was 91.3% (432/473).

In terms of the type of birth, all the spontaneous vaginal deliveries (SVD) in S4 (104/104) were attended by midwives, and therefore, there was no TOC. In contrast, none of the S4 births in which TOC occurred were SVD (0/45). Meanwhile, in S3, 35.1% (125/356) of the deliveries that featured transfers of responsibilities were SVD and attended by obstetricians. The risk of TOC associated with labour ending in dystocia was the highest in S1 (S1, *p* < 0.001; S2, *p* < 0.001; S3, *p* < 0.001).

With respect to the condition of the perineum when no TOC occurred, the S4 registered the highest number of cases with intact perineum or 1st or 2nd degree perineal tears [94.2% (98/104)], compared with the figure of 72.3% (436/603) for S2. In contrast, when TOC occurred, the highest rate of intact perineum or 1st or 2nd degree perineal tears was found in S3 [44.9% (160/356)], while S4 displayed the lowest rate [4.4% (2/45)]. Meanwhile, these episiotomies or third- or fourth-degree tears were present in 95.6% (43/45) of the cases in S4 when TOC was performed. The lowest rate in this regard was found in S3 [55.1% (196/356]. Thus, when TOC occurs, the risk of episiotomy or third- or fourth-degree perineal tear (rather than an intact perineum or a first- or second-degree tear) was found in S4 to increase by a factor of 350 (*p* < 0.001). The risk of TOC and of episiotomy or third- and fourth-degree perineal tears was the lowest in S3 (*p* < 0.001).

In addition, significant differences were found with regard to the presence of postpartum haemorrhages and early initiation of breastfeeding initiation only in the sample as a whole. This effect could be explained by the construction of a multivariate logistic regression model (Wald test) between these variables and the rest of the covariables studied, observing that the obstetric unit size, the induced onset of labour and having an episiotomy or grade III-IV injury were associated with an increasing risk of having postpartum haemorrhage. Women who had an episiotomy or a grade III-IV injury were twice as likely to have a postpartum haemorrhage compared with women who had an intact perineum or a I-II degree tear [*OR* = 2.5; CI95%:1.4–4.4]; induction of labour is also a risk for postpartum haemorrhage [*OR* = 1.8; CI95%:1.1–3.1]. Moreover, doing skin-to-skin was associated with an increased probability of early initiation of breastfeeding onset [*OR* = 45.9; 95%CI: 28.89–72.77] (Table 3).

Furthermore, a multivariate logistic regression model was used in order to predict the variables that influenced the TOC. The related variables were OUS, parity, onset of labour, pharmacological stimulation of labour and episiotomy, with the OUS being the most influential variable (Table 4). Women in S3 have twice the probability [*OR* = 2.3; 95% CI: 1.4–3.6] of having a TOC compared to those in S4; being primiparous increases the probability of TOC by almost twice [*OR* = 1.9; 95% CI: 1.5–2.4]; inducing labour rises this risk by almost three times [*OR* = 2.9; 95% CI: 2.3–3.8] in comparison with spontaneous onset of labour; the use of pharmacological stimulation and epidural analgesia are also risk factors for performing TOC [*OR* = 1.3; 95% CI: 1.0–1.7, *OR* = 1.7; 95% CI: 1.2–2.4, respectively] and performing an episiotomy increases the risk of TOC by five times [*OR* = 5.3; 95% CI: 4.3–6.6]. The model obtained a percentage prediction of 73.4%.

## 4. Discussion

This cross-sectional study is part of a broader evaluation of maternity services in Spain and a centre in Ireland. This paper focuses specifically on the intrapartum TOC of low-risk women between health care professionals in obstetric units and the associated clinical and organizational factors.

The majority of women whose labour care was initiated by a midwife remained in midwifery care throughout their labour and birth. However, there were statistical differences in the proportion of women transferred from midwifery care to obstetrician care according to the OUS (number of births). The hospitals with the lowest percentages of TOC were those in S1 and S4. These OUS groups had a transfer percentage of 29.4% and 31.2% respectively. Meanwhile, S3 had the highest transfer percentage (47.1%). These transfer rates brought sharply into focus the differences between midwifery and medical models of care.

Out of all women, those in S4 (the highest number of births per year) were most likely to be provided one to one individualized care by a caseload midwifery team. These cases had the lowest frequency of oxytocin stimulation, epidural analgesia, episiotomy or severe perineal damage, and emergency caesarean section, and also registered among the lowest transfer percentages. Midwifery models recognize childbirth as a physiological process which has inherent sociocultural and psychological dimensions [16]. Our findings echo the strong existing evidence that suggests that continuity of care models achieve the best outcomes. For example, women who are attended in midwifery-led continuity models of care were found to be less likely to experience regional analgesia and severe perineal trauma [6,17].

In general terms, the variables in this study showed some association with TOC, both within each OUS and in the sample as a whole. This was not the case, however, with postpartum haemorrhages, nor with early initiation of breastfeeding, for which associations were found for the sample as a whole but not for all the OUS. Logistic regression models show that for the sample as a whole, the covariables associated with postpartum haemorrhages were the OUS, the status of the perineum and the onset of labour, while the covariable associated with early initiation of breastfeeding was skin-to-skin contact between mother and infant [18]. Thus, we could conclude that these act as confounding variables for the sample as a whole.

We observed that certain factors connected to poor perinatal outcomes, such as the induction of labour, the stimulation of labour with oxytocin and the use of epidural analgesics; also tend to be associated with the TOC from the midwife to the obstetrician. This in turns leads outcomes such as dystocic birth and episiotomies or third- or fourth-degree tears. This could be explained by the high prevalence of obstetric interventions, especially in low-risk women, which trigger the need of performing a cascade of subsequent childbirth interventions [19].

We also confirmed that there is a high degree of variability in the intrapartum care given to low-risk women, as well as a high rate of interventionism in all the hospitals, results that echo the findings of other studies [20,21,22,23]. Among the cohort of women planning to give birth in Spanish obstetric units, we found considerably greater variation in intervention percentages than we would expect, and this is not explained by any known differences in maternal characteristics. Our findings show a significant association between the size of the unit, the professional that initiates the intrapartum care and the number of obstetric interventions performed; the smaller the OUS, the more likely women are to receive care from a midwife and the less likely they are to receive obstetric interventions. Our findings confirm that variations in intervention rates are not fully explained clinical characteristics of women planning to give birth or women’s preferences. Since low-risk women have different outcomes according to the hospital where they give birth, this may reflect a more interventional practice style and woman-centred care might not be implemented or interpreted in the same way at all places [24,25].

The OUS was also the variable that appeared to be most influential for women to have a TOC above parity, onset of labour, pharmacological stimulation of labour, and episiotomy. Although the predictive model is acceptable in terms of TOC prediction, the variability explained is estimated only at 37.9%. Thus, this result must be interpreted with caution. It seems reasonable that every OUS may have a different organization of care model in which the rest of the variables involved are adapted according to the type of intervention in childbirth. Nevertheless, all obstetric units should include a philosophy of supporting normal birth and women centred care [26].

On the one hand, the care in S1 (units with smallest number of births) and in S4 (caseload team in Ireland) was initiated in almost in all the cases by a midwife, and these cases had the lowest transfer rate out of all women in both countries and the highest spontaneous vaginal birth rate. On the other hand, larger OUS (S3) had the highest prevalence of a high level of oxytocin stimulation, epidural use, and instrumental births, and in these cases the majority of women were transferred to an obstetrician. These findings are in agreement with previous research that suggests that “low-risk” women are more likely to have a spontaneous vaginal birth in hospitals with smaller maternity departments or in midwifery-led units that operate a policy geared towards normal birth [25,27,28]. However, the findings differ with earlier studies that had showed mixed or inconclusive results on the relationship between unit size and intervention rates [29].

In general, the institutional factors affecting intervention rates are poorly understood. However, it is critical to come to an understanding of childbirth practices as an organizational cultural phenomenon. The culture of a given work environment may encourage care providers to take similar decisions, and variations are therefore not merely individual. Differences in perceptions and attitudes may result in differences in local practice and guidelines [30].

In Spain, midwives’ scope of practice follows European directives [31], and the Ministry of Health promotes assistance based on the best evidence available, with appropriate use of technology for avoiding unnecessary procedures. Intrapartum care is performed in hospitals staffed by teams of midwives and obstetricians. However, there is a significant lack of midwifery staffing in the country [32]. In fact, in the region of Catalonia, all hospitals have more obstetricians than midwives, with just one exception: public hospitals classified as S3 [33], which nonetheless showed the highest proportion of transfers in our study. Furthermore, we found a strong link between larger hospitals and higher numbers of spontaneous vaginal deliveries attended by obstetricians when women were transferred in Spain. Actually, there were no reported cases of women having a spontaneous vaginal birth assisted by an obstetrician in Ireland. This might also reflect the lack of midwifery staffing in Spain and a lack of autonomy in midwives’ practice in the country, especially in S3 hospitals. Transfers might be associated with the embedded social and cultural norms of an institution or their consequences on birth practices [13,33].

We would like to highlight that in S2 and S3 the percentages of induction of labour, oxytocin stimulation, severe perineal damage and episiotomy are considerably higher than in many European countries [8]. It is then plausible to assume that the elevated number of obstetric interventions and transfers of care is at least partly attributable to the medical obstetric practice that is predominant in Spain, which tends to be highly interventionist [33]. Unwarranted variation in medical care is also associated with non-evidence-based practice [24,34]. In addition, recent studies have shown some association between junior doctor staffing levels and intervention rates [35,36]. This could be interpreted as negative, as the majority of obstetricians and midwives are trained in these units.

These results should spark debate on the effect the organization of services may have in the Spanish context. All the public maternity units are technically orientated and able to attend both high risk and low-risk births. This raises issues as to the dubious benefits of giving birth in mixed environments, as recent studies have shown positive health results when healthy women are treated in midwife-led units that are separated from conventional obstetric units [6,37].

Previous research has shown that a given hospital’s policies and procedures, inadequate staffing, technology-focused care, and a lack of continuity of care are all barriers to a more humanized approach to birth at specialized hospitals [38]. The findings here, therefore, suggest that the current model of maternity care in these Spanish hospitals should be reconsidered in light of the impact these practices may have on maternal and neonatal outcomes. In addition, expanding midwife-led maternity services for eligible women may offer a means of reducing costs compared to the current leading model of care [39,40].

### 4.1. Strengths

This is the first study that has examined TOC and the associated clinical and organizational factors in Spain. A strength of this study is that we were able to evaluate the TOC in a homogeneous cohort of “low-risk” women that received care in different settings and health care systems in Spain and Ireland.

### 4.2. Limitations

The weakness of the existing socio-cultural studies of birth practices is that they fail to explore the organizational culture dimensions of the institutions and their role and power to bring about changes to humanize birth practices. Nonetheless, this study has limitations of its own which should be noted. The first concerns the study’s observational design. It is not possible to establish causal relationships in our study, for instance, between TOC and type of birth. Whether there is a causal relationship between these variables needs to be investigated in further research. However, the results are consistent with the findings from previous studies that showed better maternal outcomes when midwifery continuity models of care are implemented [6]. Secondly, another limitation, and one that probably applies to all research in this area, is the extremely limited available data on the characteristics of maternity units in Spain. The data we gathered on the characteristics of the included obstetric units in the study included their size and the model of care offered. However, we were unable to consider whether the women could have opted for an alternative place of birth or to look at the actual reason for TOC or the level of midwifery staffing [38], and we acknowledge this could have provided us with a more accurate interpretation of the results. These findings should therefore be interpreted with caution. Additionally, as an observational study, there is a probability that data collection might present some discontinuity. However, all investigators were encouraged to collect the data contemporaneously and continuously and signed a commitment form for that.

### 4.3. Implications for the Practise and Research

As stated in a *Lancet* series on midwifery, the available data strongly suggest an urgent need for more research to assess the appropriate interventions for childbirth [1]. Qualitative research is required to improve our understanding of the barriers and facilitating factors midwives encounter when promoting normal birth in specialized hospitals, where highly technological and medicalized birth practices exist. Our research group is currently working on identifying these in obstetric units in Catalonia, Spain. In addition, it is very important to gather the opinions of service users and professionals if we aim to provide woman-centred care [41].

Midwifery-led care should be promoted in Spain to support midwives and allow them to work autonomously to their full scope of practice [29]. Developing a national strategy taking into consideration women’s wishes and demands and the latest evidence for maternity care could be a potential approach [15]. Additionally, in-house professional development programs, including for medical and midwifery staff, are necessary to address the lack of knowledge regarding the concept of midwifery autonomy. Active involvement by midwives in groups drafting hospital guidelines and in-service development programs may also contribute to their professional development [42,43]. Finally, following national guidelines and applying recommendations of the WHO might help hospitals in Spain reduce their use of interventions for women who have a spontaneous onset of labour and minimize unwarranted variation in the use of interventions [30,38,41].

## 5. Conclusions

This exploratory study of “low-risk” births planned in obstetric units suggests that the size of the obstetric unit and the level of provision of midwifery-led care within the institution may explain some of the variation in intervention rates. To provide maternity care of optimal quality, public health stakeholders need to be aware of the childbirth practices in different organizations and then ensure that these conform to women’s and their families’ needs.

## Figures and Tables

**Figure 1 ijerph-17-08394-f001:**
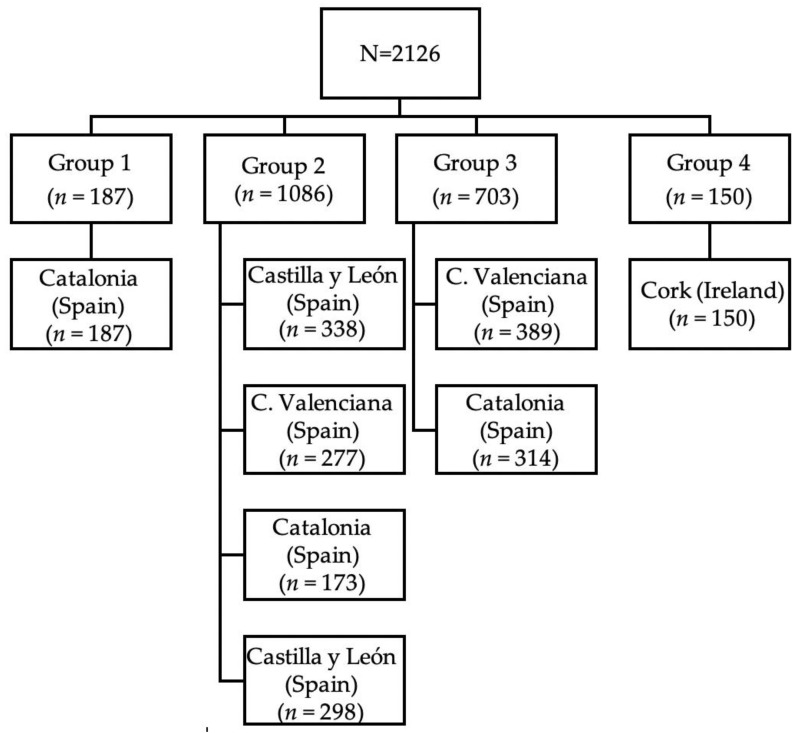
Study flow chart.

**Table 1 ijerph-17-08394-t001:** Characteristics of the sample, by obstetric unit size (*N* = 2126).

	Obstetric Unit Size									
	S1 *n* = 187		S2 *n* = 1086		S3 *n* = 703		S4 *n* = 150			
	*n*	%	*n*	%	*N*	%	*n*	%	Total	*p* *
Onset of labour										
Spontaneous	163	87.2	745	68.6	492	70.0	127	84.6	1527	<0.001
Induced	23	12.3	331	30.5	181	25.7	22	14.7	557	
C-section	1	0.5	10	0.9	30	4.3	1	0.7	42	
Pharmacological stimulation of labour										
No	129	69.0	419	38.6	208	29.6	99	66.0	855	<0.001
Yes	58	31.0	667	61.4	495	70.4	51	34.0	1271	
Epidural analgesia										
No	30	16.0	169	15.6	102	14.5	28	18.7	329	0.632
Yes	157	84.0	917	84.4	601	85.5	122	81.3	1797	
Type of birth										
Normal	145	77.5	757	69.7	442	62.9	104	69.3	1448	<0.001
Dystocic	42	22.5	329	30.3	261	37.1	46	30.7	678	
Weight of newborn (g)										
<2500	5	2.7	15	1.4	6	0.9	0	0.0	26	<0.001
2501–3000	34	18.2	227	20.9	142	20.2	10	6.7	413	
3001–3500	84	44.9	505	46.5	302	43.0	52	34.7	943	
3501–4000	49	26.2	284	26.2	212	30.2	53	35.3	598	
>4001	15	8.0	55	5.1	41	5.8	35	23.3	146	
Perineum										
No episiotomy or 1st or 2nd degree	111	59.4	555	51.1	398	56.6	100	66.7	1164	0.001
Presence of episiotomy or 3rd or 4th degree	76	40.6	531	48.9	305	43.4	50	33.3	962	
Postpartum haemorrhage										
No	182	97.3	1064	98.0	676	96.2	139	92.7	2061	0.002
Yes	5	2.7	22	2.0	27	3.8	11	7.3	65	
Early skin-to-skin contact										
Yes	180	96.3	959	88.3	646	91.9	141	94.0	1926	0.001
No	7	3.7	127	11.7	57	8.1	9	6.0	200	
Early initiation of breastfeeding										
Yes	165	88.2	823	75.8	552	78.6	124	83.2	1664	0.001
No	22	11.8	263	24.2	150	21.4	25	16.8	460	
Professional attending the onset of labour										
Midwife	181	96.8	901	83.0	546	77.7	149	99.3	1777	<0.001
Obstetrician	6	3.2	185	17.0	157	22.3	1	0.7	349	
Professional attending the birth										
Midwife	132	70.6	677	62.3	324	46.1	104	69.3	1237	<0.001
Obstetrician	55	29.4	409	37.7	379	53.9	46	30.7	889	
Birth attended by midwife from start to end										
Yes	132	70.6	603	55.5	317	45.1	104	69.3	1156	<0.001
No	55	29.4	483	44.5	386	54.9	46	30.7	970	

* Chi-square test; S1 = Obstetric Unit Size 1 (<600 births per year); S2 = Obstetric Unit Size 2 (from 601 to 1200 births per year); S3 = Obstetric Unit Size 3 (1201 to 2400 births per year); S4 = Obstetric Unit Size 4 (>2400 births per year).

**Table 2 ijerph-17-08394-t002:** Transfer of care distribution between different studied variables, and Odds Ratio and 95% confidence intervals (*N* = 2084).

	Obstetric Unit Size																								
	S1					S2					S3					S4					Total				
	Midwife Start to End				*p **	Midwife Start to End				*p **	Midwife Start to End				*p **	Midwife Start to End				*p **	Midwife Start to End				*p **
	Yes (132)		No (54)			Yes (603)		No (473)			Yes (317)		No (356)			Yes (104)		No (45)			Yes (1156)		No (928)		
	*n*	% col	*n*	% col		*n*	% col	*n*	% col		*n*	% col	*n*	% col		*n*	% col	*n*	% col		*n*	% col	*n*	% col	
Onset of labour																									
Spontaneous	120	90.9	43	79.6	0.048	502	83.3	243	51.4	<0.001	263	83	229	64.3	<0.001	97	93.3	30	66.7	<0.001	982	47.1	545	26.2	<0.001
Induced	12	9.1	11	20.4		101	16.7	230	48.6		54	17	127	35.7		7	6.7	15	33.3		174	8.3	383	18.4	
Pharmacological stimulation																									
None	107	81.1	21	38.9	<0.001	288	47.8	121	25.6	<0.001	141	44.5	65	18.3	<0.001	90	86.5	8	17.8	<0.001	626	0.3	215	10.3	<0.001
Yes	25	18.9	33	61.1		315	52.2	352	74.4		176	55.5	291	81.7		14	13.5	37	82.2		530	25.4	713	34.2	
Epidural analgesia																									
None	28	21.2	2	3.7	0.003	128	21.2	41	8.7	<0.001	79	24.9	23	6.5	<0.001	25	0.24	3	6.7	0.013	260	12.5	69	3.3	<0.001
Yes	104	78.8	52	96.3		475	78.8	432	91.3		238	75.1	333	93.5		79	0.76	42	93.3		896	0.43	859	41.2	
Type of birth																									
Normal	132	100.0	13	24.1	<0.001	603	1	154	32.6	<0.001	317	1	125	35.1	<0.001	104	1	0	0	<0.001	1156	55.5	292	0.14	<0.001
Dystocic	0	0.0	41	75.9		0	0	319	67.4		0	0	231	64.9		0	0	45	1		0	0	636	30.5	
Episiotomy																									
None vs. I-II grade	100	75.8	10	18.5	<0.001	436	72.3	118	24.9	<0.001	236	74.4	160	44.9	<0.001	98	94.2	2	4.4	<0.001	870	41.7	290	13.9	<0.001
Yes vs. III-IV grade	32	24.2	44	81.5		167	27.7	355	75.1		81	25.6	196	55.1		6	5.8	43	95.6		286	13.7	638	30.6	
Postpartum haemorrhage																									
None	129	97.7	52	96.3	0.584	595	98.7	459	0.97	0.06	308	97.2	341	95.8	0.337	98	94.2	40	88.9	0.252	1130	54.2	892	42.8	0.01
Yes	3	2.3	2	3.7		8	1.3	14	0.03		9	2.8	15	4.2		6	5.8	5	11.1		26	1.2	36	1.7	
Early skin-to-skin contact																									
Yes	128	97.0	51	94.4	0.411	528	87.6	424	89.6	0.289	297	93.7	322	90.4	0.122	98	94.2	42	93.3	0.833	1051	50.4	839	40.3	0.239
None	4	3.0	3	5.6		75	12.4	49	10.4		20	6.3	34	9.6		6	5.8	3	6.7		105	0.05	89	4.3	
Early initiation of breastfeeding																									
Yes	118	89.4	46	85.2	0.42	449	74.5	367	77.6	0.234	242	76.6	289	81.2	0.144	85	82.5	39	86.7	0.529	894	42.9	741	35.6	0.047
None	14	10.6	8	14.8		154	25.5	106	22.4		74	23.4	67	18.8		18	17.5	6	13.3		260	12.5	187	0.09	

*p* *: Chi-squared test; S1 = Obstetric Unit Size 1 (<600 births per year); S2 = Obstetric Unit Size 2 (from 601 to 1200 births per year); S3 = Obstetric Unit Size 3 (1201 to 2400 births per year); S4 = Obstetric Unit Size 4 (>2400 births per year); CI, Confidence Interval; *ns*, non-significant value > 0.05.

**Table 3 ijerph-17-08394-t003:** Multivariate logistic regression between haemorrhage, breastfeeding and related covariables (*N* = 2084).

		*p*-Value	*OR*	CI 95% for EXP (B)	
				Lower	Upper
Haemorrhage	S4 (Ref)	0.001			
	S1	0.045	0.3	0.1	0.9
	S2	0.000	0.2	0.9	0.4
	S3	0.014	0.4	0.9	0.8
	Induced	0.028	1.8	1.1	3.1
	Episiotomy	0.001	2.6	1.5	4.4
	Constant	0.001	0.1		
Breastfeeding	Skin-to-skin contact	0.001	45.9	28.9	72.8
	Constant	0.001	0.2		

**Table 4 ijerph-17-08394-t004:** Multivariate logistic regression of variables related to TOC (*N* = 2084).

	*p*-Value	*OR*	95% CI *OR*	
			Lower	Upper
Unit Size				
S4 (Ref)	0.001			
S1	0.498	0.8	0.5	1.4
S2	0.273	1.3	0.8	2.0
S3	0.001	2.3	1.5	3.6
Parity				
Multiparous (Ref)				
Nulliparous	0.001	2	1.6	2.4
Beginning of delivery				
Spontaneous (Ref)				
Induction	0.001	3	2.3	3.8
Pharmacologic stimulation				
None (Ref)				
Yes	0.011	1.4	1.1	1.8
Epidural analgesia				
None (Ref)				
Yes	0.001	1.7	1.2	2.4
Episiotomy				
None (Ref)				
Yes	0.001	5.3	4.3	6.6
Constant	0.001	0.1		

Nagelkerke R^2^ = 0.379; percentage prediction = 73.4%.

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
