# Peer review of "Transfers of Care between Healthcare Professionals in Obstetric Units of Different Sizes across Spain and in a Hospital in Ireland: The MidconBirth Study"

_ijerph, 2020, doi:10.3390/ijerph17228394_

Round 1

Reviewer 1 Report

Thank you for giving me the opportunity to review the manuscript entitled „

 Transfers of Care between Healthcare Professionals in Obstetric Units of Different Sizes: the  MidconBirth Study”.

Please find below my specific comments which in my opinion can improve the paper:

  • In my opinion the sentence in line 34-36 in the abstract can be deleted. “The Chi-square test……”
  • The sentence „This article is part of the MidconBirth study.” Needs to be rewritten – I think that the article is not the part of MidconBirth study but it presents the results obtained within MidconBirth study.
  • The sentence in lines 100-105 is very long and difficult to follow – could you please try to make it more simple.
  • There are 3 hospitals in Spain and 1 in Ireland – is it possible that they are different regarding other factors not only unit size. Maybe the organization or procedures are different considering that they are located in different countries.
  • The definition of TOC is not given in the methods part of the manuscript – I suppose it is the same as stated in background part of the paper – maybe some reference need to be repeated (or at least “as defined above”).
  • Secondary outcomes need to be defined/described.
  • I would add the information about the regions to the hospitals in  figure 1 (without this it is difficult to compare the text with figure). The authors can consider inclusion of figure 1 as supplementary materials.
  • US group – looks strange as US is mostly dedicated to the United States of America. Maybe the authors can change it into OUS groups - obstetric unit size -as it is used also in the paper – please make it uniform
  • I would add the first subheading into the results part of the manuscript (characteristics of the population or basic description…) and . Transfer analysis would be 3.2

Author Response

Thank you for your comments.

Reviewer 2 Report

Comments:

  1. I think it would beneficial to give statistics about c-sections at the end of the fist paragraph, which have increased dramatically over the past several years.
  2.  Is there a citations to support this statement: "In general, midwives tend to be responsible for women 88 with low-risk pregnancies during labour."?
  3. The introduction ends abruptly without a clear set up of the case for the contribution of this study of even what its main research questions are. The introduction is the weakest portion of the manuscript.
  4. No justification is given for choosing regions from Spain for this study that uses data from across Europe.
  5. There are lots of outcomes without laying out a key rationale for "why these particular outcomes"?
  6. Table 2 is quite busy. It seems that t-tests would be better suited to this table rather than OR. The OR makes it seem like it might be adjusted for covariates but I do not believe they are. This is not clear. 
  7. The conclusion is the strongest part of the paper. However, there needs to be much more time dedicated to the limitations of the study. 

Author Response

Thank you for your comments.

Round 2

Reviewer 1 Report

The authors have improved the paper according to my comments.

Reviewer 2 Report

The authors did an satisfactory job addressing concerns.